# Sestrins as a Therapeutic Bridge between ROS and Autophagy in Cancer

**DOI:** 10.3390/cancers11101415

**Published:** 2019-09-22

**Authors:** Miguel Sánchez-Álvarez, Raffaele Strippoli, Massimo Donadelli, Alexandr V. Bazhin, Marco Cordani

**Affiliations:** 1Mechanoadaptation & Caveolae Biology Lab, Cell and Developmental Biology Area, Centro Nacional de Investigaciones Cardiovasculares (CNIC), 28029 Madrid, Spain; miguel.sanchez@cnic.es; 2Department of Molecular Medicine, Sapienza University of Rome, 00161 Rome, Italy; raffaele.strippoli@uniroma1.it; 3Gene Expression Laboratory, National Institute for Infectious Diseases “Lazzaro Spallanzani” IRCCS, 00161 Rome, Italy; 4Department of Neurosciences, Biomedicine and Movement Sciences, Section of Biochemistry, University of Verona, 37134 Verona, Italy; 5Department of General, Visceral and Transplantation Surgery, Ludwig-Maximilians University, 81377 Munich, Germany; 6German Cancer Consortium (DKTK), Partner Site Munich, 80366 Munich, Germany; 7IMDEA Nanociencia, C/Faraday 9, Ciudad Universitaria de Cantoblanco, 28049 Madrid, Spain

**Keywords:** Sestrins, ROS, autophagy, stress response, nutrient management, cancer therapy

## Abstract

The regulation of Reactive Oxygen Species (ROS) levels and the contribution therein from networks regulating cell metabolism, such as autophagy and the mTOR-dependent nutrient-sensing pathway, constitute major targets for selective therapeutic intervention against several types of tumors, due to their extensive rewiring in cancer cells as compared to healthy cells. Here, we discuss the sestrin family of proteins—homeostatic transducers of oxidative stress, and drivers of antioxidant and metabolic adaptation—as emerging targets for pharmacological intervention. These adaptive regulators lie at the intersection of those two priority nodes of interest in antitumor intervention—ROS control and the regulation of cell metabolism and autophagy—therefore, they hold the potential not only for the development of completely novel compounds, but also for leveraging on synergistic strategies with current options for tumor therapy and classification/stadiation to achieve personalized medicine.

## 1. ROS in Health and Disease

### 1.1. Regulation of ROS Levels in Physiology

The term “Reactive Oxygen Species” (ROS) encompasses a group of highly reactive chemical entities containing molecular oxygen, including oxygen radicals (i.e., superoxide (O_2_^•−^), and hydroxyl (^•^OH), peroxyl (RO_2_^•^), and alkoxyl (RO^•^) radicals) and non-radicals (i.e., hypochlorous acid (HCIO), singlet oxygen (^1^O_2_) and hydrogen peroxide (H_2_O_2_)). These molecules are by-products—even under physiological conditions—mainly from redox transactions taking place as part of the respiratory activity of the OXPHOS system in mitochondria, as well as NADPH oxidases [1].

The regulation of ROS levels poses an important logistic problem for metazoan cells. Primarily powerful oxidizing by-products of metabolism, uncontrolled ROS species can damage different structures in the cell (lipids, proteins) compromising its function and viability, and jeopardizing the accurate transmission of genetic information by damaging genomic material and increasing its mutation rates. However, ROS also emerge as pivotal intra- and extracellular messengers, encoding information about the functional/metabolic state of different structures for the regulation of numerous signaling pathways and the tuning across orders of magnitude of different limiting metabolic nodes, changing cell sensitivity to other environmental conditions, such as fuel availability or usage [2]. Thus, the control of ROS levels is paramount for cell homeostasis, and their dysregulation (either by excess or defect) is both a recurring cause and common consequence of disease states. 

In normal conditions, intracellular ROS levels are tightly controlled by intricate networks of antioxidant molecules, as the glutathione pair (GSSG/GSH) or the nicotinamide adenine dinucleotide pair (NADH/NAD^+^); and associated redox enzymes, as superoxide dismutases (SODs), catalase, glutathione peroxidases (GPXs), peroxiredoxins (PRXs) or thioredoxins (TRXs). These as yet poorly understood networks are integrated with signaling pathways and functional modules that directly impact ROS levels, as is the case of protein folding routes in the endoplasmic reticulum [3], or the superoxide dismutase activities in the mitochondria [4]. Due to their profound and broad potential impact on cell homeostasis, these redox effectors in turn are integrated with and regulated by robust homeostatic systems, such as the Unfolded Protein Response (UPR) [5,6] and the related Integrated Stress Response (ISR) [7,8], genotoxic responses [9,10], and programs ensuring adaptation to nutrient deprivation [11,12] (see below Section 1.3). 

### 1.2. A Prominent Example of the Cell Conundrum: ROS in Cancer Biology

Increased ROS levels were consistently observed for most tumor types at least two decades ago [13] and were interpreted as both a by-product of increased metabolism in tumor cells, as well as an oncogenic agent because of its damaging and mutational properties. Apart from increased net metabolic rates and proliferation, tumor cells rewire different routes of carbon usage, such as anaplerosis and enhanced glycolytic flux, to supply building blocks for the increased demand of lipid, amino acid and nucleotide anabolism [14,15]. These pathways can be both sources of antioxidant NAD(P)H molecules, but also sources of ROS by-products, and represent a first source of specificity and synergy when intervening ROS levels in cancer cells [15]. The combined targeting of tumor cell metabolism and ROS management is therefore an appealing source of intervention selectivity. Indeed, cancer cells do often “relax” oxidative stress surveillance mechanisms, a most prominent example being the p53 tumor suppressor [16], an event that may simultaneously drive dysregulated signaling (see below Section 1.3). It must be noted that, in contrast with the wild type protein, mutant p53 isoforms fail to exert antioxidant activities and rather increase intracellular ROS, favoring cancer progression in parallel with different anabolic programs, as part of their Gain-of-function (GOF) phenotype (see below Section 1.3) [17]. 

For this reason, challenging ROS homeostasis in cancer cells may represent both an Achilles’ heel *per se* in tumor cells, but also a robust opportunity for synergistic strategies [9,10]. Emerging avenues of research in this regard are the leveraging on differential ascorbate toxicity, which has been proven effective against aggressive tumor cell types with very elevated metabolic rates, such as glioblastoma [18,19]; and the exploitation of the phenomenon known as ferroptosis [16,20]. Both strategies, highly interrelated, rely on the limited availability of the glutathione antioxidant system and high sensitivity to uncontrolled H_2_O_2_ accumulation in tumor cells. 

Interestingly, ROS also emerge as drivers of pro-tumorigenic survival through signaling rewiring. Thus, paradoxically, tumor cells are relatively exposed to ROS-mediated damage, but nonetheless benefit from increased ROS levels to promote the activation of signaling networks that drive increased growth and proliferation. For example, reduced ROS production rates resulting from dampened mitochondrial function upon depletion of mitochondrial transcription factor A (TFAM), blunt pro-survival and proliferative signaling in different experimental models, such as Notch and β-catenin signaling in mouse keratinocytes [21] and driving K-Ras-dependent signaling in lung cancer [22]. Supporting this concept, invasive and metastatic phenotypes in different tumor cells of B16F10 melanoma and super-invasive human cervix cancer are distinguished from poorly metastatic ones by enhanced production of mitochondrial superoxide [23].

### 1.3. Dysregulated Nutrient/Growth Homeostasis Signaling as a Source of ROS

A key aspect of ROS-related cancer biology is the contribution of dysregulated systems determining energy and nutrient homeostasis in the cell. Mechanistic Target Of Rapamycin Complex 1 (mTORC1) and 5′ adenosine monophosphate-activated protein kinase (AMPK) interpret multiple cues, including oxidative stress, to integrate them with the control of energy management, anabolism and cell growth. Not surprisingly, their dysregulation is pervasive in tumor cells of virtually any kind [24,25].

mTORC1 is one of the major signaling nodes coupling environmental and metabolic signals such as nutrients, growth factors, oxygen, and stress with the control of protein synthesis, lipid anabolism and cell growth [26]. The mTORC1 complex comprises different subunits—mTOR/FRAP1 kinase, regulatory-associated protein of mTOR (Raptor), mammalian lethal with SEC13 protein 8 (MLST8), proline-rich Akt substrate of 40 kDa (PRAS40) and DEP domain-containing mTOR-interacting protein (DEPTOR)—and integrates two major limiting signal inputs. mTORC1 is a central target in growth factor signaling, through its derepression upon phosphorylation of the tuberous sclerosis complex subunits TSC1 and 2 and activation of the positive regulator Rheb. However, although growth factors drive mTORC1 activation, amino acids are required for it to be complete [27]. Amino acid availability is monitored by the active Rag complex—consisting of the small GTPases RagA:B in the GTP-bound state and RagC:D in the GDP-bound state (RagA:BGTP-RagC:DGDP)—and promotes mTORC1 translocation to the lysosome by interacting with the mTORC1 specific subunit Raptor. This allows for the full activation by GTP-bound Rheb of the mTOR kinase [28]. A crucial determinant of the activation state of Rag small GTPases, of relevance to the focus of this review, is the “GATOR” super-complex, composed of two complexes of opposing function: GATOR1 (containing DEP Domain Containing 5 (DEPDC5), Nprl2 and Nprl3, negatively regulating RagA:B, and considered an important tumor suppressor [29]); and GATOR2 (a positive regulator of the Rag complex composed of Mios; the WD Repeat Domain proteins WDR59 and WDR24; Seh1L and Sec13) [29].

The AMP-activated protein kinase (AMPK) is a ubiquitous heterotrimeric Ser/Thr kinase complex that functions as part of an energy-sensing pathway in eukaryotic cells. AMPK is regulated by adenylate levels in the cell (i.e., ATP, ADP and AMP) [30]. Enhanced AMP:ATP ratios—either as a result of reduced production or increased consumption of ATP—activate AMPK through phosphorylation at its threonine 172 residue [31]. AMPK activation promotes energy conservation and cell survival by activating signaling pathways of catabolic metabolism and/or inhibiting anabolic processes consuming ATP. Importantly, AMPK is also a key hub of a tumor suppressor network regulating cell growth and proliferation in response to stress [31]. In this context, AMPK is pivotal for sensing and buffering mitochondrial ROS, conferring stress resistance and metabolic homeostasis [32].

AMPK is activated by phosphorylation by the tumor suppressor liver kinase B1 (LKB1) [33], whereas the tumor suppressors Tumor Suppressor Complex 2 (TSC2) [34] and p53 [35] are downstream targets of AMPK activity. Interestingly, p53 may in turn increase AMPK activity through transcriptional activation of the gene encoding the β subunit of the enzymatic complex [36] and Sestrins [37] (see below Section 3.1), nucleating a positive feedback that sustains AMPK signaling. However, some studies report that gain-of-function GOF p53 mutants can instead inhibit AMPK phosphorylation in response to metabolic stress, sustaining an oncogenic signaling network that favors tumor progression [38,39].

It must be noted that, apart from the phenotypic outputs themselves being major ROS sources, these pathways feed, in a concerted fashion, with other homeostasis surveillance pathways such as the Unfolded Protein Responses (UPRs), into a central hub controlling ROS levels and ROS-dependent cell regulation: autophagy.

## 2. Autophagy and Cancer

### 2.1. Macroautophagy

The term “autophagy” refers to a group of regulated mechanisms by which cells break down specific building blocks and structures, enabling their recycling, the rerouting of their energy, or their disposal when they are damaged and/or toxic [40]. While the list of pathway variants has grown steadily in recent years, the core machinery is rather defined and evolutionarily conserved. The Unc51-like kinases ULK1/2 are major phosphoregulated initiator switches of the pathway, and their activating dephosphorylation elicits the activation of the Beclin-1/Vps34/Atg14L-containing complex, which triggers the nucleation of the phagophore- the precursory membrane structure from which the autophagic vesicle is formed by phosphorylation of phosphatidylinositol (PtdIns) into PtdIns(3)P [41]. This lipid signal then recruits the WIPI2-Atg16L modification complex, which catalyzes the covalent conjugation of the ubiquitin-like protein Atg12 to Atg5. The resulting complex activates Atg3, which covalently attaches mammalian homologues of the ubiquitin-like yeast protein Atg8 (also collectively known as Atg8ls; the most studied being LC3A-C, GATE16 and GABARAPL1-3) to phosphatidylethanolamine (PE) on the surface of autophagosomes. PE-conjugated Atg8ls drive the maturation and closure of autophagosomes, enabling the docking of specific cargos and adaptor proteins such as Sequestosome-1/p62. These proteins regulate the fusion of the autophagosome with the lysosomal compartment as mediated by multiple proteins, including SNAREs and UVRAG [42]. This summarizes the canonical autophagy cascade, which is primarily linked to pathways sensing nutrient and energy availability in the cell through the phosphorylation-mediated control of the ULK kinases.

### 2.2. Dual Role of Autophagy in Cancer

Autophagy plays pivotal roles across both physiological phenomena, such as development and postnatal growth, as well as a variety of human diseases, including cancer [43]. The modulation of autophagy is a recurrent trait in cancer cells, as may be expected from its tight regulation by nutrient and energy sensing pathways such as AMPK and mTOR, so frequently altered in cancer [44]. Importantly, the impact of autophagy in cancer cell death or survival is highly contextual and dependent on tumor origin, type and stage; and the environmental context tumor cells lie in (for example, well-irrigated tumors, versus poorly perfused, starving and hypoxic tumors) [45].

Increasing evidence supports the hypothesis that autophagy might protect cancer cells from further damage as a result of their increased metabolism, by eliminating impaired organelles or recycling misfolded macromolecules [46], especially in the conflicting context of poorly perfused, hypoxic tumors, where metabolism and proliferation are not properly curbed in accordance to limited nutrient availability [45]. Thus, by increasing stress tolerance and providing an alternative nutrient source by which cancer cells can satisfy their massive nutrient and energy demands, autophagy likely plays an oncogenic role for tumor cells. In this context, autophagy appears as an appealing therapeutic strategy as its inhibition exposes these cancer cell types to the accumulation of damage and apoptosis [47].

However, uncontrolled autophagy can trigger death-type II, likely due to excessive degradation of cellular constituents and organelles required for homeostasis of cells. Furthermore, autophagy opposes different anabolic programs in the cell and the accumulation of cell mass. Thus, its *potentiation* can also have a tumor-suppressive effect. Cancer cells evading this suppressive activity from autophagy usually exhibit a malignant phenotype, with enhanced ROS production and increased genomic instability [48]. Defects in autophagy cause the accumulation of abnormal mitochondria, which are a potential source of ROS that lead to genomic instability, and cancer initiation and progression [49]. Moreover, autophagy defects also activate the DNA damage response and promote genetic instability and inflammation, which in turn sustains tumorigenesis [50].

This evidence strongly suggests that autophagy is a key determinant of tumor initiation, behavior and survival.

## 3. Sestrins at the Crossroads of ROS, Oxidative Stress and Autophagy

### 3.1. The Sestrin Protein Family

Sestrins (SESNs) are a family of proteins that are induced upon various conditions of stress, DNA damage, hypoxia and metabolic alterations [51]. SESNs are highly conserved across taxa: invertebrates have a single orthologue, while vertebrates have three paralogues—SESN1-3—these genes are located on different chromosomes: SESN1 on 6q21, SESN2 on 1p35.3, and SESN3 on 11q21 [52].

SESNs are ubiquitously expressed in adults, albeit at different levels across tissues [52,53]. Interestingly, SESN1 and SESN2 are most highly expressed in skeletal muscle [53]. SESN1 can be expressed as a short form (SESN1S; with a molecular weight of 55kDa, the most similar to SESN2), and a long-form (SESN1L, with an extended N-terminus) [54]. Different sources of stress may induce SESN expression, including DNA damage, oxidative stress, nutrient scarcity and hypoxia. Transcription factors responsible for the expression of SESNs include nuclear factor erythroid 2-like 2 (Nrf2), NH (2)-terminal kinase (JNK)/c-Jun pathway and hypoxia-inducible factor-1α (Hif-1α). Whereas SESN1 and SESN2 are mainly responsive to p53 [52,53], SESN3 is activated by forkhead box O (FoxO) transcriptional factors. SESN3 can also be regulated by serum and cytokines/growth factors [55]. While information in this regard is still limited, a number of reports directly link dysregulated SESN expression levels with specific types of cancer cells [56,57,58], suggesting a potential for classifying and inferring oxidative stress resilience phenotypes.

Understanding SESN structure and its functional implications has been challenging, since these proteins do not show obvious similarity with any known structural domain or catalytic motif [51]. The determination of hSESN2 structure by X-ray crystallography revealed that hSESN2 bears two structurally similar subdomains, SESN-A and SESN-C, connected by a helix-loop-helix domain (SESN-B). Both subdomains share significant structural homology with proteins belonging to the alkylhydroperoxidase family, responsible for catalyzing the reduction of peroxiredoxins, such as *M. tuberculosis* AhpD [59,60]. Specific sites in these domains are relevant for the interaction of sestrins with other proteins and functional modulation: for example, a catalytic cysteine (C125) and conserved surrounding residues (Y127 and H132) in SESN-A are critical for the antioxidant function of hSESN2, and an aspartate-aspartate (DD) motif embedded in the C-terminal domain is required for the interaction between hSESN2 and GATOR2 (see below Section 3.3). Moreover, a leucine-binding site is present in the SESN-C domain, suggesting that hSESN2 might act as a direct sensor of leucine availability [61]. Finally, SESN2 binds kelch-like ECH-associated protein 1 (Keap1), the autophagy adapter p62/sequestosome-1 (SQSTM1) and the autophagy initiator ULK1 [62]. These interactions underscore the involvement of SESN2 in the linkage of antioxidant responses with autophagy regulation.

### 3.2. Sestrins as Master Regulators of ROS Management

SESN expression is induced upon oxidative stress and is strictly associated to the antioxidant response (Figure 1). Transcription factors induced during oxidative stress such as p53, Nrf2, AP-1 and FoxOs may control SESN expression. Notably, SESN1 expression is induced by hydrogen peroxide in a strictly p53-dependent manner, whereas the induction of SESN2 by oxidative stress is only partially dependent on p53 activation [63]. The role of SESN in antioxidant defense is demonstrated by the fact that the antioxidant response is impaired after SESN silencing of all three known isoforms [55,64], whereas it is enhanced after SESN ectopic expression. However, although this antioxidant function of SESNs is well established, the biochemical basis of this activity is not fully clear yet.

An oxidoreductase active site of SESN-A containing a catalytic cysteine and other conserved residues was found to be critical for the antioxidant function of hSESN2. Although structural and biochemical evidence demonstrates that hSESN2 has an intrinsic peroxidase activity, the physiological ROS substrates of hSESN2 have not been identified so far. Importantly, the antioxidant activity of SESN is not likely derived from a direct reduction and regeneration of the catalytic active site of other antioxidant effectors, such as peroxiredoxins (Prx) [65]. It was hypothesized that SESN may promote the activity of other oxidoreductases, such as sulfiredoxin (Srx), that also regenerate Prxs. Indeed, SESNs can increase Srx expression through activation of Nrf2, a transcription factor that controls the expression of a range of antioxidants [62].

Independently of their Prx-regulating activity, SESNs may contribute to redox homeostasis through negative regulation of the mTORC1 signaling pathway [66]. mTOR activity may enhance ROS production via inhibition of autophagy or directly acting on mitochondrial function.

### 3.3. Sestrins and Nutrient Sensing: An Entry Point to Intervene Autophagy?

Several studies reported that SESNs play a critical role as sensors of amino acid changes in the extracellular environment, which is a key signal in the regulation of the mTORC1 pathway (Figure 2). In this regard, SESN2 induction is required for cell survival during glutamine deprivation [67]. Some studies revealed that a SESN-dependent and AMPK-independent mechanism for mTORC1 inhibition might be mediated by the interaction of SESN2 with GATOR2, which directly binds at least this SESN paralog [54,59,68]. This interaction results in the suppression of lysosomal mTOR localization in a Rag-dependent manner and represents a mechanism by which SESNs potentially integrate responses to oxidative damage with equalization of mTORC1 activation threshold and nutrient sensing in the cell [54,59,68]. SESN2 might also act as a direct sensor of leucine independently from that interaction [60,61,69]. Intriguingly, according to another report, SESN1, but not SESN2 exerts control on mTORC1 activity after leucine treatment in skeletal muscle [70]. At present, we do not fully understand the relationship between these two links to amino acid availability sensing and their respective contextuality, and further studies are needed to clarify the wiring of SESN2 to nutrient homeostasis. Additional mechanisms have been proposed in this regard. A master regulator of amino acid sensing, General Control Non-derepressible 2 (GCN2), positively regulates SESN2 through the translational derepression of its downstream effector ATF4. This link bears particular interest as it represents a potential access for GCN2 activity to the control of mTORC1 activation [71]. It must be noted that this same axis links the control of SESN expression and activity to other stress responses—Unfolded Protein Responses from mitochondria and endoplasmic reticulum; proteotoxic responses operated by the EIF2AK4/HRI kinase; and the pathogen-sensing EIF2AK2/PKR kinase—converging on the Integrated Stress Response centralized on the phosphorylation of the Ser51 residue of the eIF2a translation initiation factor. Finally, in response to leucine starvation, SESN2 can be phosphorylated by ULK1, thus affecting mTORC1 activation in the fashion of a classical positive feedback loop, conferring robustness to a nutrient stress and autophagy response network [72].

AMPK inhibits mTORC1 through the phosphorylation of TSC2 and Raptor in response to cellular energy cues [73]. A number of studies reveal that SESNs-dependent AMPK induction is important for mTORC1 suppression in diverse cellular contexts [74,75,76,77]. SESN1 and SESN2 were shown to suppress mTOR signaling through the activation of AMPK and TSC2 phosphorylation in a p53-dependent manner upon DNA damage [37]. In response to genotoxic stress, SESN2 expression is upregulated favoring sustained AMPK activity by sustaining the recruitment of LKB1. Moreover, both AMPK and SESN2 coordinate the suppression of Akt-mTOR signaling after ionizing radiation in breast cancer cells [78]. Notably, SESN3 enhances hepatic insulin sensitivity through the positive regulation of mTORC2-Akt signaling [79]. The main differences between Sestrins are summarized in Figure 2.

These findings and other evidence support the hypothesis that SESNs might play an important role as a hub integrating the monitoring of nutrient availability and antioxidant and proteotoxic stress responses, to control major growth signaling and energy management networks (i.e., mTORC1 and AMPK).

## 4. Sestrins and the Therapeutical Management of Cancer

As discussed above, SESNs are key antioxidant proteins that control metabolism and energy stresses through a sophisticated regulation of the mTORC1/AMPK signaling pathway. Hence, SESNs are potential routes of intervention to profoundly alter their metabolic and redox homeostasis, rendering them more sensitive to chemotherapy drugs. The development of novel chemicals and biotechnological compounds to modulate SESN expression and function for the treatment of cancer and other diseases are thus an active field of research (see Figure 3).

Here we provide an overview on recent advances regarding the contribution of SESN expression modulation to the pharmacological effects of different compounds and the identification of novel means to drive such modulation, with a focus on anticancer therapy research. We also provide a Table summarizing this literature (Table 1).

### 4.1. Modulation of SESNs by Natural Compounds

Many purified compounds from dietary sources have been investigated for their anticancer activities through a direct modulation of SESN expression [80,81,82,83,84,85,86,87,88].

Quercetin (3,3′,4′,5,7-pentahydroxyflavone) is a polyphenolic compound with preventive effect against colorectal and pancreatic cancer [100]. Interestingly, quercetin reduces mitochondrial membrane potential, sustains intracellular ROS production and increases SESN2 expression, favoring apoptosis and cell death in HCT116 and HT-29 colon cancer cells. Thus, SESN2 might represent an effective mediator of the cytotoxic activity of quercetin through the regulation of mTOR and AMPK/p38 pathways in a p53-independent manner [80,81].

Cucurbitacin B a tetracyclic triterpenoid belonging to the family of cucurbitacins, exhibits several anti-proliferative and pro-apoptotic effects in cancer and enhances the chemotherapeutic effects in vitro and in vivo cancer models [101,102,103,104]. Of note, cucurbitacin B treatment also increases SESN3 at both mRNA and protein levels in epidermal growth factor receptor (EGFR)-mutant lung cancer cells, and this upregulation significantly contributes to the anti-proliferative and apoptotic effect of cucurbitacin B in EGFR-mutant lung cancer cells, providing a mechanistic rationale for its attractive potential therapeutic use in non-small cell lung carcinoma (NSCLC), especially given the challenging management of therapeutic resistance typical of these tumors [82].

Eupatilin (5,7-dihydroxy-3,4,6-trimethoxyflavone) is a flavone derived from *Artemisia asiatica* exhibiting many pharmacological activities which include several antioxidant [105,106], anti-inflammatory [107,108], anti-atherogenic [109] and anti-cancer properties [110]. Eupatilin induces the expression of SESN2 protecting hepatocytes against arachidonic acid (AA) and iron-induced oxidative stress. This induction of SESN2 by eupatilin modulates autophagy-mediated protection against ferroptosis and may have potential use to prevent toxic liver damage from xenobiotics [83].

Isorhapontigenin (ISO), a new derivate of stilbene isolated from Chinese herb *Gnetum cleistostachyum*, induced autophagy in human bladder cancer cells, contributing to the inhibition of anchorage-independent growth of cancer cells. Interestingly, SESN2 expression was dramatically downregulated in tumors and ISO-mediated autophagy induction occurred in a SESN2-dependent and BECN1-independent manner [84]. The upregulation of SESN2 expression after ISO treatment occurred via MAPK8-Jun-dependent transcriptional regulation [84], which provides a novel mechanistic insight into the cytotoxic effect of ISO on bladder cancers. This study suggests that ISO might act as a promising preventive and/or therapeutic drug against human bladder cancer by modulating SESN2 expression.

Arsenic trioxide (ATO) is a traditional Chinese medicine having chemotherapeutic effects against many cancers [111,112,113,114,115,116,117]. Although the molecular mechanisms underpinning its cytotoxic effect remain poorly characterized, ATO-induced oxidative stress is believed to cause defects in genomic stability, epigenetic modulation, gene expression, cell cycle progression, and in triggering the mitochondrial pathway of apoptosis [118,119]. Puzzlingly, ATO can also promote antioxidant activities that can compromise its desired effects in a context-dependent manner [120]. Liang-Ting et al. reported that ATO suppresses miR-182-5p expression in several cancer cell lines, correlating with the upregulation of SESN2 mRNA [85]. The pre-treatment with the free radical scavenger N-acetyl-cysteine (NAC) reduced oxidative stress and ATO-mediated suppression of miR-182-5p as well as the associated enhancement of SESN2 expression. Importantly, supply of miR-182-5p-mimicking molecules significantly suppressed the expression of SESN2 and was associated with longer survival of glioma or lung cancer patients [85]. This example underscores the importance of considering these types of relationships when designing therapeutic approaches in relation to the modulation of SESN2.

An intriguing target for study is the liver X receptor-α (LXRα), a member of the nuclear receptor superfamily of ligand-activated transcription factors, that serves as a lipid sensor and master positive regulator of lipid metabolism in the liver [121]. Sterol regulatory element binding protein-1c (SREBP-1c) is an important target gene of LXRα activation [122] and is a key transcription factor of lipogenic gene expression [123] The LXRα-SREBP-1c axis is an attractive target for the prevention and/or treatment of lipid anabolism networks both driving hepatic steatosis and supporting the high rates of growth and proliferation in different forms of cancer [124,125].

Resveratrol (3,4′,5-trihydroxystilbene) is a natural polyphenolic component with anti-oxidant, anti-tumor and anti-inflammatory activities [126,127]. However, it is not known whether resveratrol affects LXRα-dependent lipogenic gene expression [128], So Hee et al. observed that resveratrol inhibited LXRα-dependent activation of SREBP-1c, and thereby inhibited target gene expression in hepatocytes in a manner independent from AMPK and Sirt1. Importantly, SESN2 expression strongly increases after treatment with resveratrol and repressed LXRα and SREBP-1c transcriptional throughput expression, suggesting that SESN2 is an important effector of resveratrol-dependent modulation of those lipid anabolism networks [86].

### 4.2. Other Therapeutics Acting on SESNs Expression

Cabazitaxel is a semisyntheric taxane currently undergoing both clinical trials in patients with metastatic castration-resistant prostate cancer (CRPC) and further preclinical characterization, on the basis of its poor affinity for ATP-dependent drug efflux pump P-glycoprotein [129,130,131]. Cabazitaxel treatment in several CRPC cell lines exerted higher cytotoxic effects as compared to other taxanes, such as docetaxel and paclitaxel. This chemotherapy activity elevated ROS production through the downregulation of SESN3, supporting an opportunity for synergistic blockade of SESN3 in taxane-based therapies [89].

Proteasome inhibitors and ER stress-inducing drugs like bortezomib and nelfinavir resulted in the upregulation of SESN2 along with ER stress markers in breast, ovarian and cervical adenocarcinoma cancer cell lines [90]. Interestingly, ectopic expression of the UPR PERK branch effector ATF4 transcriptionally upregulates of SESN2, which in turn contributes to two hallmarks of a sustained ER stress response: mTORC1 inhibition and autophagy. Interestingly, cancer cells treated with the nelfinavir showed reduced mTOR activity and increased ATF4 and SESN2 expression levels. As nelfinavir and bortezomib are currently being tested in clinical trials of patients with solid cancers, the identification of the mTOR-inhibitory activities of these drugs may help to better explore their chemosensibility-modulating effects.

Several histone deacetylase inhibitors (HDACIs) have been characterized in detail, including suberoylanilide hydroxamic acid (SAHA), trichostatin A (TSA), and depsipeptide [132]. Notably, HDACIs exhibit potent antitumor activities, exerted through the induction of cell cycle arrest, differentiation, apoptosis and cell death in a variety of cancer cells [133,134]. Recent studies have demonstrated that HDACIs, such as SAHA and TSA, also induce autophagy in human cancer cells [135,136], but the molecular mechanisms behind HDACIs-mediated autophagy are not clear yet. The forkhead box protein 1 (FOXO1) is transcription factor that plays a pivotal role in autophagy induction [132,137,138,139]. While the role of FOXO1 in the HDACIs-induced autophagy has not been directly demonstrated and characterized, HDACIs induce protective autophagy by sustaining the expression and transcriptional activity of FOXO1 in HCT116 colon cancer cells and HepG2 hepatoma cells. Interestingly, FOXO1-mediated autophagy is largely contributed by the suppression of mTORC1 via transcriptional upregulation of the SESN3 gene [91]. The discovery of a novel function of FOXO1 in HDACIs-mediated autophagy in human cancer cells may support the development of a novel combinatorial anticancer therapeutic strategies leveraging on a potential synergy between HDACIs and inhibitors of SESNs. Besides HDACIs, the inhibition of the other epigenetic regulators such as histone demethylase LSD1 can promote autophagy through the direct control of SESN2 expression in certain physiopathological conditions, such as during inflammation and in neuroblastoma [140,141]. Moreover, genetic and pharmacological inhibition of the methyltransferase EZH2 controls SESN1 expression and its subsequent activity on mTORC1 in follicular lymphoma [142,143].

Chronic myeloid leukemia (CML) and Ph+ acute lymphoblastic leukemia (ALL) are characterized by the presence of the BCR-ABL oncoprotein, which leads to activation of a plethora of mitogenic and pro-survival pathways, including the mTOR signaling cascade [144,145]. In BCR-ABL expressing cells, treatment with tyrosine kinase inhibitors (TKIs) upregulates SESN3 expression both at the level of mRNA and protein. Importantly, overexpression of SESN3 results in curbing of mTORC1 activity and exerts potent proliferation suppression activity on leukemic cells expressing either WT-BCR-ABL or mutant TKI-resistant BCR-ABL, arguing for a promising potential in antileukemic strategies [92].

PTEN/MMAC1 (phosphatase and tensin homolog) has been identified as a tumor suppressor gene for a variety of cancers [146]. However, PTEN is frequently deleted in cancer and its inactivation results in the constitutive activation of the PI3K-AKT pathway with a subsequent increase of protein synthesis, cell cycle progression, invasiveness and survival to anoikis [147]. Following exposure to DNA-damaging agent topotecan, ATM serine/threonine kinase phosphorylates PTEN at Ser113 which, in turn, mediates its nuclear translocation. Importantly, PTEN nuclear translocation promotes autophagy through activation of the p-Jun-SESN2-AMPK pathway in A549 and HeLa cells tumor cells [93], supporting SESNs as potential synergistic targets in therapies against PTEN-null tumors.

Genetic factors that influence inflammation and energy production in cells may affect patient outcomes following treatment with external beam radiation therapy (EBRT). A study conducted in patients affected by prostate cancer reported that downregulation of SESN3 gene was associated with fatigue intensification during EBRT [94]. Inactivation of SESN3 might lead to dysregulation of signaling pathways such as mTOR-AMPK, resulting in mitochondria impairment and oxidative stress. Thus, SESN3 may serve as an interventional target and a biomarker for cellular and molecular events associated with EBRT-related fatigue.

These and other studies reported that SESNs modulation by chemotherapeutic drugs and/or radiotherapy may affect cancer cells survival regulating signaling pathways related to oxidative stress, mitochondria damage and apoptosis [88,94,95,96,97,98,99]. Therefore, SESNs may represent valid markers with prognostic value in cancer cells.

## 5. Conclusions

Due to their intersection with two priority mechanisms for anticancer therapy—ROS levels and autophagy—and their structural and functional peculiarities, sestrins appear as attractive novel pharmacological targets. The contextuality of their function also hints at their potential for synergistic strategies and personalized medicine. Moreover, modulation of sestrin-dependent signaling is also emerging as a field of active interest for the therapeutic control of toxicity and metabolic imbalance, two potential precursors to oncogenic transformation in oncological challenges such as liver and pancreatic cancer [148,149,150]. Future work will assess the true therapeutic potential of these novel players in cancer biology.

## Figures and Tables

**Figure 1 cancers-11-01415-f001:**
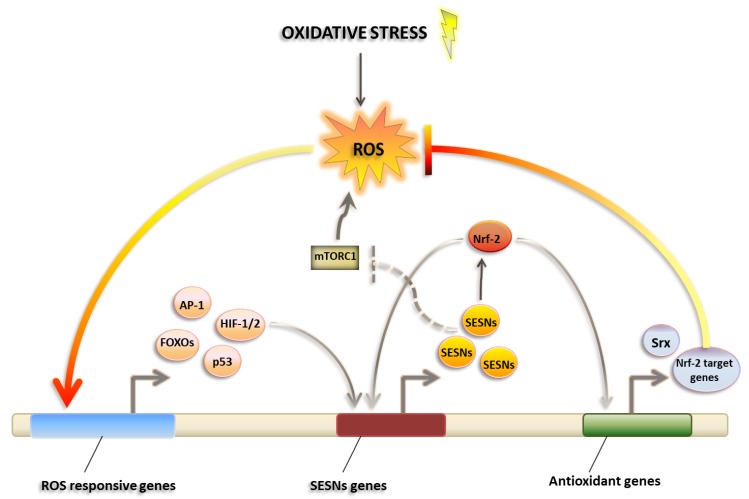
Sestrins as master regulators of Reactive Oxygen Species (ROS) management. Sestrins expression is induced upon oxidative stress and leads to antioxidant response. Increased ROS levels in cells activate ROS-responsive genes including transcription factors AP-1, FOXOs, HIF-1/2 and p53, which in turn induce the transcription of SESNs genes. SESNs coordinate antioxidant responses by activating Nrf-2, a transcription factor leading expression of Nrf-2 target antioxidant genes. In addition, SESNs orchestrate ROS homeostasis by inhibiting mTORC1, leading to oxidative stress.

**Figure 2 cancers-11-01415-f002:**
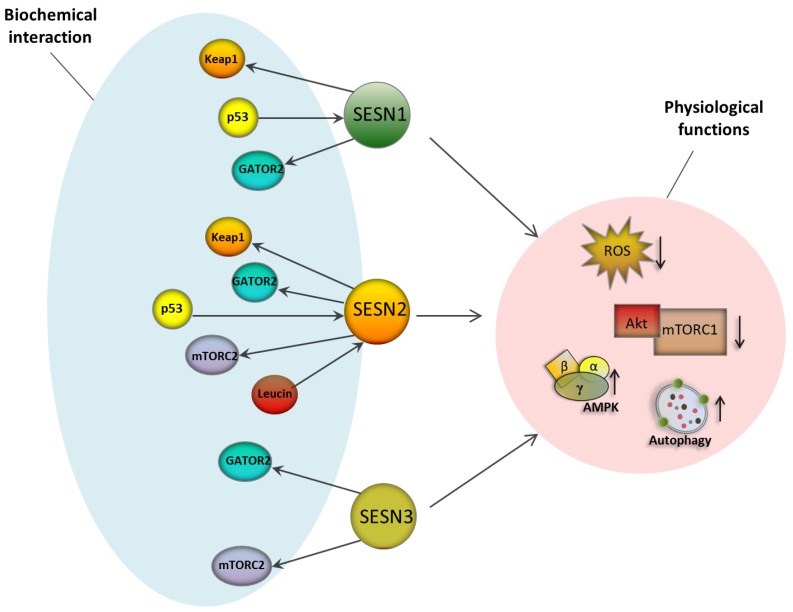
Similarity and differences between members of the sestrin family. SESN1, SESN2 and SESN3 activate antioxidant and pro-autophagic responses by modulating mTORC1 and AMPK signaling pathways, as well as through a positive impact on major antioxidant effectors such as Srxs and Prxs (through mechanisms poorly identified) [37,56,64]. However, some differences between SESNs occur at a biochemical level. Indeed, some of them directly interact with specific effectors. SESN1 and SENS2 (but not SESN3) interact with Keap1 to promote its autophagic degradation [62]. All three SESNs interact with GATOR2 to negatively regulate mTORC1 [54,62,68]. SESN1 and SESN2 are activated by p53 to modulate mTOR signaling [37,64,66]. SESN2 and SESN3, but not SESN1 interact physically with mTORC2 to modulate Akt signaling [79]. SESN2 interacts with leucine to negatively regulate mTORC1 [79].

**Figure 3 cancers-11-01415-f003:**
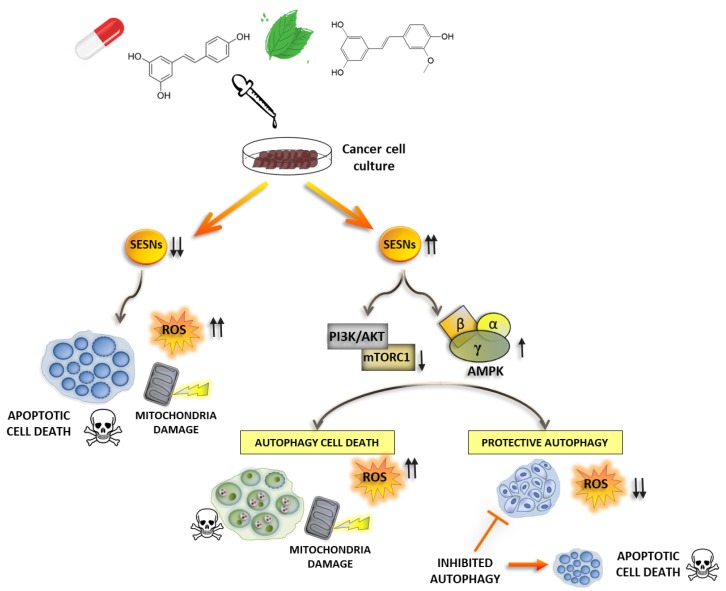
Sestrins (SESNs) as a therapeutical management of cancer. Natural compounds or synthetic therapeutics lead to changes in SESNs expression in a manner dependent on the type of drug, the target tissue and the metabolic status of cancer cells. Enhanced SESNs expression might result in autophagy activation, leading alternatively to cell death or survival in cancer cells, depending on the differentially induced autophagic program. Some drugs may also reduce sestrins expression in cancer cells, leading to ROS enhancement, mitochondria damage and apoptotic cell death.

**Table 1 cancers-11-01415-t001:** Major current cancer treatments modulating SESNs expression.

Entry	Therapeutic Approach	Cell Lines	Type of Cancer	Effect on SESNs	Molecular Mechanisms	Biological Effect	Refs
1	Quercetin	HCT116, HT-29	Colon cancer	SESN2 ↑	mTOR ↓, AMPK/p38 ↑	ROS, apoptosis, cell death	[80,81]
2	Cucurbitacin B	A549, H1792, H1650, H1975	Lung cancer	SESN3 ↑	PI3K/mTOR ↓, STAT-3 ↓ AMPKα ↑	Anti-proliferative, apoptosis	[82]
3	Eupatilin	HepG2	Liver cancer	SESN2 ↑	Autophagy and antioxidant genes ↑	Protective autophagy, reduction ROS, hepatoprotection	[83]
4	Isorhapontigenin	UMUC3, T24T, HeLa	Bladder, cervix cancer	SESN2 ↑	MAPK8-Jun	Autophagy, inhibition of cell growth	[84]
5	Arsenic trioxide	U87MG, patients derived glioma cells, A549, H1299	Glioma, lung cancer	SESN2 ↑	miR-182-5p ↓	Antioxidant response, increased patient survival	[85]
6	Resveratrol	HepG2	Liver cancer	SESN2 ↑	LXRα-SREBP-1c ↓	Inhibition of hepatic lipogenesis	[86]
7	Carnosol	HCT116, SW480	Colon cancer	SESN2 ↑	PERK/Nrf2/SESN2↑	Reduction of cell viability, apoptosis,	[87]
8	Tanshinone IIA	43B, MG63	Osteosarcoma	SESN2↑	MAP4K4 SAPK/JNK1/Jun kinase↑, Jun recruitment to AP-1-binding site in the SESN2 promoter region, PI3K/Akt ↓	Anchorage-independent growth inhibition; osteosarcoma progression; mitochondrial dysfunction, Autophagy induction	[88]
9	Cabazitaxel	C4-2AT6	Prostate cancer	SESN3 ↓	Cleaved-PARP ↑	ROS, citotoxicity	[89]
10	Bortezomib, Nelfinavir	MDA-MB-453, OVCAR3, HeLa	Breast, ovarian, cervix cancer	SESN2 ↑	ATF4, ATF3, CHOP ↑ mTOR ↓	Autophagy, ER stress, Proteasome inhibition	[90]
11	Suberoylanilide hydroxamic acid, trichostatin A, depsipeptide	HCT116, HepG2	Lung, liver cancer	SESN3 ↑	FOXO1 ↑, mTOR ↓	Protective autophagy	[91]
12	Tyrosine kinase inhibitors	BV173, BV173R, Ba/F3 p210T315I, U937, KT-1	Leukemia	SESN3 ↑	mTORC1 ↓	Antileukemic response	[92]
13	Topotecan	A549, HeLa	Lung, cervix cancer	SESN2 ↑	PTEN nuclear translocation, p-Jun-SESN2-AMPK ↑	Autophagy	[93]
14	External beam radiation therapy		Prostate cancer	SESN3 ↓	AMPK-mTORC1↓	Mitophagy, Oxidative Stress, Fatigue intensification during EBRT.	[94]
15	Carbonyl cyanide m-chlorophenyl hydrazine (CCCP)	SH-SY5Y	Neuroblastoma	SESN2 ↑ (early time) SESN2 ↓ (prolonged exposure)	RBX1 mediated ubiquitination	Protection from mitochondrial damage	[95]
16	2-imino-6-methoxy-2H-chromene-3-carbothioamide (IMCA)	TT	Thyroid cancer	SESN1 ↑, SESN2 ↑	AMPK ↑ mTORC1 ↓	Inhibition of cell proliferation, Apoptosis induction	[96]
17	3,4,5,4′-tetramethoxystilbene (DMU-212)	A-2780, SKOV-3	Ovarian cancer	SESNs ↑	P53 signaling ↑	Apoptosis induction, Inhibition of cell proliferation, reduction of tumor growth in vivo	[97]
18	Ultraviolet radiations (UVA, UVB)	NHEM, iMC23	Melanoma	SESN2 ↑	P53 and AKT3 pathway	Inhibition of UVB-induced DNA damage repair; Promotion of UVA-induced ROS generation	[98]
19	ChlA-F	RT4, T24T, UMUC3	Bladder cancer	SESN2 ↑	Autophagy signaling ↑	Anchorage-independent growth inhibition	[99]

Table summarizing literature on chemotherapeutics targeting expression levels and/or function of SESNs in different tumor cell lines and cancer types. Impact on either aspect of SESN biology is denoted as ↑ (increased SESN expression or increased downstream function) and ↓ (decreased SESN expression level or attenuated downstream function). Specific consequences for tumor cell survival is indicated when available.

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
