# Peer review of "Sestrins as a Therapeutic Bridge between ROS and Autophagy in Cancer"

_cancers, 2019, doi:10.3390/cancers11101415_

Round 1
Reviewer 1 Report
This study provides new information regarding the role of Sestrins as a therapeutic bridge between ROS and autophagy in cancers, which is mostly lacking in published studies. However, there are couple of concerns for this paper.
1. Since the title of this paper "Sestrins as a therapeutic bridge between ROS and autophagy in cancer" emphasizes Sestrins as a therapeutic bridge between ROS and autophagy, the authors are encouraged to include one or two sentences in the abstract emphasizing this bridge. Also, inclusion of a subtitle with related content discussing this bridge is encouraged to make a complete story for this paper.
2. Many of the references are in the wrong format, including 2, 3, 5-23, 28-34, 49, 50, 55-58, 64, 65, 72, 82, 85, 111, 112, 121, 127, 132, 133, 136-144.
Author Response
#Reviewer 1
This study provides new information regarding the role of Sestrins as a therapeutic bridge between ROS and autophagy in cancers, which is mostly lacking in published studies. However, there are couple of concerns for this paper.
We are grateful for the positive appreciation from reviewer 1 of our work and its potential impact.
Since the title of this paper "Sestrins as a therapeutic bridge between ROS and autophagy in cancer" emphasizes Sestrins as a therapeutic bridge between ROS and autophagy, the authors are encouraged to include one or two sentences in the abstract emphasizing this bridge. Also, inclusion of a subtitle with related content discussing this bridge is encouraged to make a complete story for this paper.
We now include changes to the abstract text emphasizing this bridge.
As to the “subtitle”, we agree it is a desirable idea, but we were unsure regarding how to proceed in terms of formatting for publication and whether a limited text would end up being redundant with our changes to the abstract. We nonetheless would propose to the editors and reviewer 1 the following subtitle:
Sestrins emerge as novel regulators of both ROS and autophagy, two deeply interdependent, pivotal aspects of cancer cell physiology. As such, Sestrins potentially constitute a convenient therapeutic target to consider for the personalized design of anticancer strategies leveraging on the intervention of both ROS and autophagy.
Many of the references are in the wrong format, including 2, 3, 5-23, 28-34, 49, 50, 55-58, 64, 65, 72, 82, 85, 111, 112, 121, 127, 132, 133, 136-144.
We now include all references in the appropriate format.
Reviewer 2 Report
The authors integrated well a lot of information about SESNs, ROS, and autophagy in cancers. The quality of review is high and beneficial for future investigations about those fields. The figures and the table are well arranged, helping readers' understanding. Some minor points should be re-considered.
Although SESN1 has been less understood, the authors should describe properties and knowledge of SESN1.
Line 64, Distinguish is a verb. Does it mean "distinctive"?
Line 77, increase is italic. On purpose?
Line 150-156, Some words are written with different font.
Line 187, 286, and 420, headlines are written in capitals. In line 36 and 139, they are expressed using lower case letters. They seem to have less concordance.
Line 179, potentiation is italic.
Line 229-230, an antioxidant activity of SESN such as the reduction of peroxiredoxins (Prxs).... Prxs shoud have antioxidant effects. Therefore, reduction of Prxs are not antioxidant.
line 266, This sentence needs a period?
Line 331, the underlying molecular mechanisms underlying its.... I think that former underlying is not necessary.
Line 392, BCRABL. In line 393, BCR-ABL.
Author Response
#Reviewer 2
The authors integrated well a lot of information about SESNs, ROS, and autophagy in cancers. The quality of review is high and beneficial for future investigations about those fields. The figures and the table are well arranged, helping readers' understanding. Some minor points should be re-considered.
We sincerely appreciate the positive comments from reviewer 2 to several aspects of our manuscript.
Although SESN1 has been less understood, the authors should describe properties and knowledge of SESN1.
Our knowledge on SESN1 is less complete as compared to what is known, for example, about SESN2. In our revised manuscript, we aim at specifically bringing this fact to light, and to elaborate on the knowledge about SESN1 and questions of interest regarding its potential contrasts with the other two members of the family throughout the text.
Line 64, Distinguish is a verb. Does it mean "distinctive"?
We apologize for this grammatical error. Indeed, we aimed at meaning ‘distinctive’. We have nonetheless changed the word altogether.
Line 77, increase is italic. On purpose?
We think reviewer 2 refers to line 80 in the previous version of the manuscript, now 84. Yes, it is on purpose, to stress the fact that Gain-Of-Function mutant p53 proteins often do not orchestrate antioxidant activities, and instead they promote ROS production.
Line 150-156, Some words are written with different font.
Indeed, some gene names copied from our data lists kept an Arial font. We have corrected that in our manuscript (lines 163 through 169 in the revised version).
Line 187, 286, and 420, headlines are written in capitals. In line 36 and 139, they are expressed using lower case letters. They seem to have less concordance.
We thank the reviewer for pointing out this important defect in formatting. All headlines are now written in capitals.
Line 179, potentiation is italic.
We thank reviewer 2 for this comment on this specific formatting, in line 192 of the revised manuscript version. The font is correct, as we aimed at conveying the idea of exacerbating autophagy as a potential opposite impact on cancer cell survival, as compared to its prosurvival basal activation, discussed in the previous paragraph.
Line 229-230, an antioxidant activity of SESN such as the reduction of peroxiredoxins (Prxs).... Prxs shoud have antioxidant effects. Therefore, reduction of Prxs are not antioxidant.
We sincerely apologize for the ambiguous use of the word ‘reduction’, which may have misled reviewer 2 to interpret it as ‘downregulation’ or ‘decreased expression’. We have reworded this sentence (lines 247 through 250 in the revised manuscript) to convey clearly the idea that “evidence supports that sestrins are not likely exerting their antioxidant role through the reversal to a reduced state of the catalytic active site of peroxirredoxins. (see old ref. 62, now listed as 65)”.
line 266, This sentence needs a period?
We thank the reviewer for pointing out the missing period symbol (in the galley proofs, this line is 275; in the revised manuscript is 289).
Line 331, the underlying molecular mechanisms underlying its.... I think that former underlying is not necessary.
We sincerely thank reviewer 2. This is a blatant typo we apologize for. We have reworded the sentence to render it correct (now line 381).
Line 392, BCRABL. In line 393, BCR-ABL.
We thank reviewer 2 for its observation. We now corrected the nomenclature as BCR-ABL in both lines (lines 448 through 4449 in the revised version of the manuscript).
We sincerely appreciate the zeal of reviewer 2. His/her comments have been enormously helpful to confer consistency to the nomenclature of the text, which has been revised to the best of our capabilities.

Reviewer 3 Report
Cancers #........
Title: Sestrins as a therapeutic bridge between ROS and autophagy in cancer
Authors: Alvarez Raffaele Strippoli , Massimo Donadelli*, Alexandr V. Bazhin * and Marco Cordani
In this review, the authors clarify Sestrins-mediated network in ROS-autophagy and metabolic adaptation. The manuscript explores, in an interesting way, how these factors are functionally and physically linked and discusses the readout of these interactions in tumor, along with potential therapeutic interventions.
The MS addresses important issues in Sestrins-ROS-autophagy biology, and I recommend publication with minor concerns.
To be addressed to a wider target of people, I suggest to the authors to present a figure (or Table) addressing the differences and similarities between Sesn1-2 and 3. The presence of three Sestrins often induced some confusions in non-specialized readers.
In section 4.2., two recent works: (Ambrosio S, et al., Oncogene. 2017 36(48):6701-6711. doi: 10.1038/onc.2017.267. Lysine-specific demethylase LSD1 regulates autophagy in neuroblastoma through SESN2-dependent pathway. Zhuo, x et al. Life Sci. 2019:116696. doi: 10.1016/j.lfs.2019.116696. Knockdown of LSD1 meliorates Ox-LDL-stimulated NLRP3 activation and inflammation by promoting autophagy via SESN2-mesiated PI3K/Akt/mTOR signaling pathway) present evidences for a potential role of the histone demethylase LSD1 in the appropriate control of SESN2 expression. Because LSD1 inhibitors are currently used as epi-drugs, I suggest the author to report these findings and discuss the role of histone de-methylases such as LSD1 in SESN2 regulation.
Author Response
#Reviewer 3
In this review, the authors clarify Sestrins-mediated network in ROS-autophagy and metabolic adaptation. The manuscript explores, in an interesting way, how these factors are functionally and physically linked and discusses the readout of these interactions in tumor, along with potential therapeutic interventions.
The MS addresses important issues in Sestrins-ROS-autophagy biology, and I recommend publication with minor concerns.
We sincerely thank reviewer 3 for his positive reading of our manuscript and its scope, and for his/her sincere recommendation for publication.
To be addressed to a wider target of people, I suggest to the authors to present a figure (or Table) addressing the differences and similarities between Sesn1-2 and 3. The presence of three Sestrins often induced some confusions in non-specialized readers.
We agree with reviewer 3 on the convenience of summarizing that information, visually if possible. We now present a graphical scheme in figure 2 integrating basic aspects of our current knowledge on the sestrin protein family.
In section 4.2., two recent works: (Ambrosio S, et al., Oncogene. 2017 36(48):6701-6711. doi: 10.1038/onc.2017.267. Lysine-specific demethylase LSD1 regulates autophagy in neuroblastoma through SESN2-dependent pathway. Zhuo, x et al. Life Sci. 2019:116696. doi: 10.1016/j.lfs.2019.116696. Knockdown of LSD1 meliorates Ox-LDL-stimulated NLRP3 activation and inflammation by promoting autophagy via SESN2-mesiated PI3K/Akt/mTOR signaling pathway) present evidences for a potential role of the histone demethylase LSD1 in the appropriate control of SESN2 expression. Because LSD1 inhibitors are currently used as epi-drugs, I suggest the author to report these findings and discuss the role of histone de-methylases such as LSD1 in SESN2 regulation.
We most sincerely appreciate this generous and constructive review from referee 3. We now include these recent references and discuss them in the text, in the section on potential novel opportunities for the therapeutic leverage on SESNs.

Round 2
Reviewer 1 Report
As this paper shows merits in the related research area, the English writing needs some improvement. For example:
L8, "due to their extensive rewiring in cancer as compared to healthy cells" should be corrected to "due to their extensive rewiring in cancer cells as compared to healthy cells". L66, "A prominent example of a prominent cellular conundrum: ROS in cancer biology" should be corrected to "A prominent example of the cellular conundrum: ROS in cancer biology". L262, "SESN1, and not SESN2 seems to exert control..." should be corrected to "SESN1, but not SESN2, seems to exert control...". L295, "autophagy degradation" should be corrected to "autophagic degradation". L316, "depending on the differential autophagic program induced" should be corrected to "should be corrected to "depending on the differentially induced autophagic program".
Author Response
We would like to take the chance to express our gratitude to reviewer 1 for his/her generosity and depth of reading of the manuscript. We have followed the corrections he/she kindly pointed out and thoroughly revised, to our best capability and with the assistance of an UK-native speaker, the grammar and spelling used.
